# Quantitative Analyses of Biofilm by Using Crystal Violet Staining and Optical Reflection

**DOI:** 10.3390/ma15196727

**Published:** 2022-09-28

**Authors:** Ryuto Kamimura, Hideyuki Kanematsu, Akiko Ogawa, Takeshi Kogo, Hidekazu Miura, Risa Kawai, Nobumitsu Hirai, Takehito Kato, Michiko Yoshitake, Dana M. Barry

**Affiliations:** 1National Institute of Technology (KOSEN), Suzuka College, Suzuka 510-0294, Mie, Japan; 2Faculty of Medical Engineering, Suzuka University of Medical Science, Suzuka 510-0293, Mie, Japan; 3National Institute of Technology (KOSEN), Oyama College, Oyama 323-0806, Tochigi, Japan; 4National Institute for Materials Science (NIMS), Tsukuba 305-0044, Ibaraki, Japan; 5Department of Electrical & Computer Engineering, Clarkson University, Potsdam, NY 13699, USA; 6STEM Laboratory, State University of New York, Canton, NY 13617, USA

**Keywords:** biofilms, crystal violet, optical reflection, color analyses, XYZ color plane, L*a*b* color plane

## Abstract

Biofilms have caused many problems, not only in the industrial fields, but also in our daily lives. Therefore, it is important for us to control them by evaluating them properly. There are many instrumental analytical methods available for evaluating formed biofilm qualitatively. These methods include the use of Raman spectroscopy and various microscopes (optical microscopes, confocal laser microscopes, scanning electron microscopes, transmission electron microscopes, atomic force microscopes, etc.). On the other hand, there are some biological methods, such as staining, gene analyses, etc. From the practical viewpoint, staining methods seem to be the best due to various reasons. Therefore, we focused on the staining method that used a crystal violet solution. In the previous study, we devised an evaluation process for biofilms using a color meter to analyze the various staining situations. However, this method was complicated and expensive for practical engineers. For this experiment, we investigated the process of using regular photos that were quantified without any instruments except for digitized cameras. Digitized cameras were used to compare the results. As a result, we confirmed that the absolute values were different for both cases, respectively. However, the tendency of changes was the same. Therefore, we plan to utilize the changes before and after biofilm formation as indicators for the future.

## 1. Introduction

A biofilm (BF) is a thin film of material formed by bacterial activity on the surface of a material or other interface. The fundamental concepts of biofilms have been clarified and explained by many researchers and summarized in some books, reviews, etc. [1,2,3,4,5]. In most cases, they form on materials’ surfaces. Therefore, biofilm formation must be affected by materials’ surfaces. From this viewpoint, we have tried to show how materials affect biofilm formation and growth [6,7]. BF is composed of about 80% water, EPS (extracellular polymeric substances), and bacteria. It has a characteristic sliminess. This sliminess is said to be caused by quorum sensing, a phenomenon in which bacteria adhere to the surface of material, multiply, and expel polysaccharides outside of the colony. In addition to polysaccharides, proteins, lipids, and nucleic acids (DNA and RNA) are produced in BF, which collectively is called EPS (Figure 1).

Bacteria in BFs have different properties from those of ordinary airborne bacteria. This affects various fields, such as medicine, environmental science, architecture, mechanics, chemical and pharmaceutical engineering, pharmaceuticals, and materials science. To control these effects, BF must be accurately evaluated by taking appropriate measures.

As described above, the appropriate evaluation method for BF is very important and the main premise for the following development of anti-biofilm materials. The evaluation methods are mainly classified into two types. One of them is the evaluation group composed of many versatile analytical instruments. These include various microscopes, such as optical microscopes, electron microscopes, confocal laser microscopes, etc., or various analytical facilities, such as Raman spectroscopy, FT-IR spectroscopy, etc. The other type is the biological evaluation group that is composed of gene analyses, staining methods, etc. These two types are combined appropriately to produce new advanced analytical methods. Examples include electron microscopes [8,9,10,11,12], confocal laser microscopes [13,14,15,16,17], IR measurements [18,19,20,21], and Raman spectroscopy [22,23,24,25,26,27,28,29]. Some proposed methods have made great contributions for clarifying the biological essence of biofilms and their relationship with materials and environments. These methods provided us with qualitive, semi-quantitative, and quantitative analyses for our research projects. However, we still need other new evaluation methods for practical applications. Practical applications mean that researchers, engineers, and general users (facing practical industrial or daily life problems) could use them to check biofilms quantitatively as well as qualitatively, and above all, products that have relatively large and unsteady shapes should be analyzed directly. In such a case, the evaluation method requires swiftness and simplicity. From the practical viewpoint, the measurement condition should be close to satisfy those requirements as much as possible. To satisfy the purpose, the SIAA (Society of International Sustaining Growth for Antimicrobial Articles, Japan, Tokyo), composed of more than 1000 Japanese companies in the antimicrobial materials field, are going to establish an ISO and we expect that it would be valid until March 2023. In this method, crystal violet staining [30,31,32] biofilms are extracted into sodium dodecyl sulfate (SDS) solution and the absorbance by 590 nm light is defined as the quantity of biofilms. However, if the stained colors of specimens could be evaluated directly, the process would be simpler. Therefore, we carried out some experiments as trials to determine the biofilm quantity by measuring surface color at the stained biofilms, so that the newly proposed method would lead to the modified quantification method in the future.

## 2. Experimental Section

### 2.1. Substrate Specimens

In this experiment, commercially available PE (polyethylene sheet), and pure Titanium specimens were used as substrates. Thin sheets (0.5 to 1.0 mm thick) of each material were cut into 10 × 10 mm^2^ pieces using metal shears and they were cleaned with alcohol. We used two specimens because we wanted to confirm the applicability of the proposed method in this experiment to both metallic materials and polymeric substances.

### 2.2. Bacteria

*Escherichia coli* (*E. coli*, K12 G6) were used as model bacteria in this study. The bacteria were selected due to the following two reasons. First, the model bacteria for this study should have low risk and should be easy to deal with. Next, we often used these bacteria in previous studies and have accumulated versatile data and experiences. Therefore, we used *E. coli* as our model bacteria.

### 2.3. Biofilm Formation

Biofilm formation was carried out by a static method. Luria–Bertani (LB) liquid medium (2068-75, M9T2881, Nacalai Co., Kyoto, Japan) was autoclaved at 121 °C for 15 min and *E.coli* were added in LB medium, so that the colony formation unit (CFU) per milli liter (mL) was around 1 × 10^9^ after a shaking incubation at 37 °C for 24 h. Next, the bacterial solution was put into 12 plastic wells, so that each well was filled with 1.2 mL of solution. Then, the specimens were immersed into wells for 0, 1, and 3 days at 25 °C in an incubator.

### 2.4. Raman Spectroscopy

We used Raman spectroscopy as a confirmation method to verify that biofilms were really formed. Specimens with BFs were pretreated by freeze dehydration in advance to carry out Raman spectroscopy. The freeze dehydration process is composed of two steps. One of them is the substitution of water in BFs with alcohol, and the other is vacuuming. The concrete steps are described as follows.

The aqueous solutions were adjusted so that the ratios of distilled water: ethanol (Ethanol, C_2_H_5_OH, 99.5%, Reagent Special Grade, 057-00451, APQ8101, Wako Pure Chemical Industries, Ltd., Osaka, Japan) were 7:3, 5:5, 3:7, 2:8, 1:9, 0.5:9.5, 0.2:9.8, and 0:1. Solutions of ethanol and t-butyl alcohol (tert-Butyl alcohol, 2-Methyl-2-propanol, special grade, 000-10915, G72121J, Kishida Chemical Co., Osaka, Japan) in the wells were aspirated with a dropper; the adjusted solution was added with a dropper and replaced in turn, and the wells were allowed to stand for 15 min. After alcohol displacement, the samples were frozen in a freezer and vacuumed using a vacuum pump.

Raman spectroscopy was carried out, using a Raman spectrometer (LabRAM HR Evolution, Horiba, Kyoto, Japan). A laser beam (532 nm) was irradiated onto the sample’s surface (diameter: approximately 1 µm), and the Raman shift was measured three times (N = 3) under the following conditions: −50% attenuation filter, 3 s exposure time, 5 integration times, grating: 300 gr/mm, and measurement wavelength range: 500 cm^−1^–3500 cm^−1^.

### 2.5. Color Analyses

An aqueous solution containing 0.1% crystal violet (CV) was prepared to stain specimens. The solution was used as a standard solution for ISO. This is because we have investigated some cases using the solution in the past. After bacterial solutions were removed from the wells, the CV solution was put into the well containing the sample and the immersion continued for 30 min. Then, the CV solution was removed, and pure water was poured into the wells to remove non-special absorbed CV, which was washed away from the specimens’ surfaces. Then the water was immediately removed. This washing process was repeated three times. As a result, we obtained stained specimens that correspond to the amounts of biofilm present.

To evaluate the extent of biofilm formation on specimens, the staining must be analyzed quantitatively. In usual cases, the stained parts are extracted into a proper solution and the absorbances are measured [33,34]. On the contrary, we measured the stained violet color on specimens by optical reflection, using color meters. Then, by combining three color parameters, L*, a* and b* were obtained [35]. In this study, we analyzed the color reflection of stained parts using photos and image analyses. A digital camera (1066C004, PSG7X Mark II, Canon Inc., Tokyo, Japan), a black box, and a ring light source using a white LED were set up for photographing the specimens. The camera parameters used in the shooting were aperture f = 9.0, shutter speed SS = 1/40, and ISO sensitivity 125. The photographed samples were analyzed using ImageJ, and histograms of each RGB color within the measurement range on the image were obtained. The histograms were converted into the XYZ color system (Equation (1)).
(1)$$\left(\begin{array}{c} X\\ Y\\ Z \end{array}\right)=\left(\begin{array}{ccc} 0.4124 & 0.3576 & 0.1805\\ 0.2126 & 0.7152 & 0.0722\\ 0.0193 & 0.1192 & 0.9505 \end{array}\right)$$

Then, they were converted into the Yxy color system (Equation (2)), and plotted on the xy chromaticity diagram, according to the following equations:(2)$$x=\frac{X}{X+Y+Z}\;,\;y=\frac{Y}{X+Y+Z}$$

To compare the results by this new method with those by using a color meter in the past, we measured the specimens’ stained surfaces, using a color measurement device (Color Reader, CR-13, Konica Minolta Sensing, Inc., Tokyo, Japan). The results were then plotted on an xy color diagram.

## 3. Results and Discussions

### 3.1. Confirmation of Biofilm Formation on Both Specimens

The results of the Raman spectrometer measurements of pure titanium specimens immersed in *E. coli* culture solution (for different periods of time) are shown in Figure 2. The wavenumber (cm^−1^) is on the horizontal axis and the intensity is on the vertical axis.

The results for the Ti substrate alone showed no specific peaks. On the contrary, specimens immersed in the bacterial solution of LB showed peaks at 2930 cm^−1^, 1660 cm^−1^, 1440 cm^−1^, and 1320 cm^−1^. These are typical peaks for biofilms as compared to those we previously confirmed for specimens where biofilms formed on them.

Figure 3 shows the results of the Raman spectrometer measurements of the samples immersed in *E. coli* culture solution (for different periods of time) on the PE substrate.

In the measurement for only the PE substrate, sharp peaks were detected at 2880 cm^−1^, 1440 cm^−1^, 1290 cm^−1^, 1130 cm^−1^, and 1060 cm^−1^, and a broad peak was found at around 2160 cm^−1^, respectively. Obviously, these peaks were derived from PE itself. However, we could observe that these original PE-derived peaks were clearly reduced. Furthermore, the extent of the reduction increased with the immersion time. Figure 3 shows that it was hard for us to analyze biofilm peaks because the PE-derived peaks were relatively strong. We enlarged the results for the specimen immersed in *E. coli* for 3 days. They are displayed in Figure 4.

In Figure 4, peaks at 2930 cm^−1^, 1660 cm^−1^, 1440 cm^−1^, and 1320 cm^−1^ were also observed, even though they were not so remarkable. Since they were typical peaks for specimens with biofilms, we could confirm biofilms also on the surface of PE.

### 3.2. Results of Staining Evaluations

After staining samples on Ti substrates with different immersion periods in the *E. coli* culture medium, measurements were made using a colorimeter. The results were plotted on an xy chromaticity diagram, as shown in Figure 5 below.

The dots in the upper right of the figure for the Ti substrate only shift to the lower left as the immersion period in the *E. coli* culture medium increases. The mean values and standard deviations of the colorimetric measurements of Ti substrate and Ti specimen immersed in *E. coli* culture medium for 3 days are shown in Table 1 below.

After staining the PE substrate, the samples were immersed in the *E. coli* culture solution for different periods of time. Then, they were measured by using a colorimeter and plotted on an xy chromaticity diagram (Figure 6). In the PE samples, as in the Ti samples, there is a tendency that the point cloud shifts to the lower left as the immersion period increases.

The color difference between two points (x1, y1) and (x2, y2) on the xy chromaticity diagram is defined as ΔC, according to the following equation:(3)$$\Delta C=\sqrt{\left\{{\left(x1-x2\right)}^{2}+{\left(y1-y2\right)}^{2}\right\}}\;\;$$

The color difference between the substrate and samples immersed in the *E. coli* culture medium (for 3 days) in Ti and PE, respectively, was measured using a colorimeter. Table 2 shows the color difference between the two samples.

After staining samples on Ti substrates with different immersion periods in the *E. coli* culture medium, measurements were made using an image analysis technique. The results were plotted on an xy chromaticity diagram, shown in Figure 7 below.

The mean values and standard deviations of the results of staining the Ti substrate and the samples immersed in the *E. coli* culture medium (for 3 days) are shown in Table 3.

Results for the color evaluations show that the mean values of the two methods (color meter measurements and image analyses) were different, but the standard deviation and the trend of the color change in the immersed samples were similar. The colorimetric method was the same as the image analysis method.

We started this research project to complete the evaluation method as a quantitative one. At this point, we have not completed it. However, we showed that this method has the potential to be a substitution method for the extraction one. To rapidly quantify the biofilm of products or large-scale specimens, the color change should be expressed as concrete figures, such as vector values or more statistical ones. We will continue this project to obtain our final goal.

## 4. Conclusions

In this experiment, we investigated the process where usual photos were quantified without any instruments, except for the usual digitized cameras. The results were compared by using a digitized camera. We obtained the following results from our experiments:(1)We confirmed by Raman spectroscopy that biofilms formed both on titanium and PE specimens, respectively.(2)Although the average of the number of color values obtained by the method using image analysis is different from that by the method using a colorimeter, the accuracy and trend of the shift of the point cloud are almost the same. Therefore, the method using image analysis is effective as an alternative colorimetric method to the method using a colorimeter.(3)We found that, in the future, it is possible that the image analyses from photos could be applied to the evaluation of biofilms.

## Figures and Tables

**Figure 1 materials-15-06727-f001:**
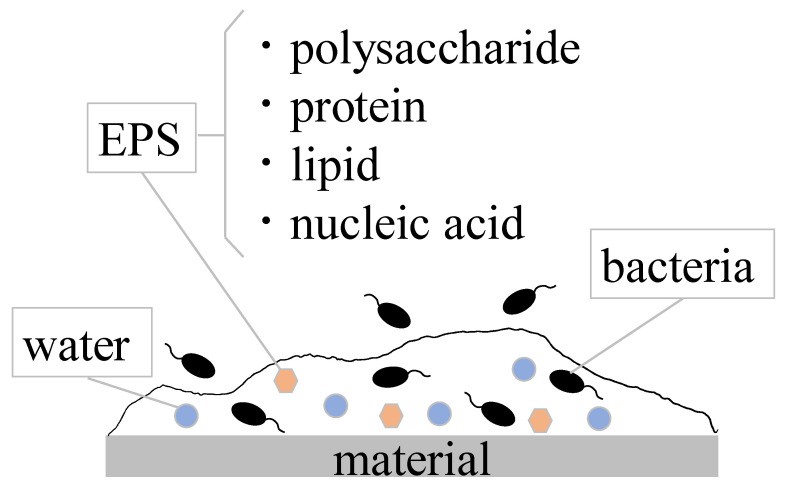
Schematic diagram of biofilms on materials.

**Figure 2 materials-15-06727-f002:**
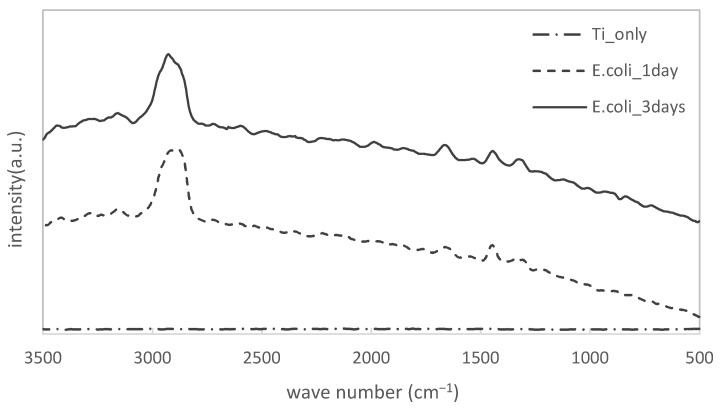
Change of Raman shifts for titanium specimens immersed in LB media filled with *E. coli*.

**Figure 3 materials-15-06727-f003:**
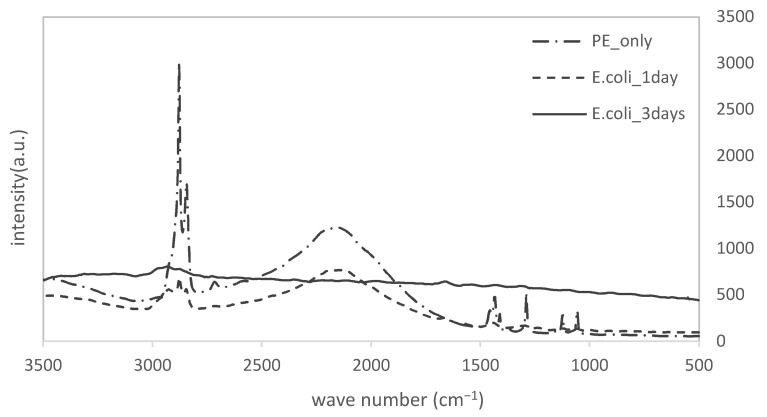
Raman shifts of PE specimens with immersion time.

**Figure 4 materials-15-06727-f004:**
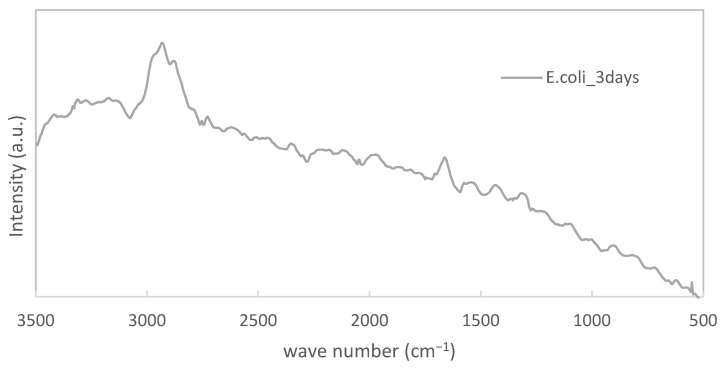
Enlarged results for PE immersed in LB bacterial solution.

**Figure 5 materials-15-06727-f005:**
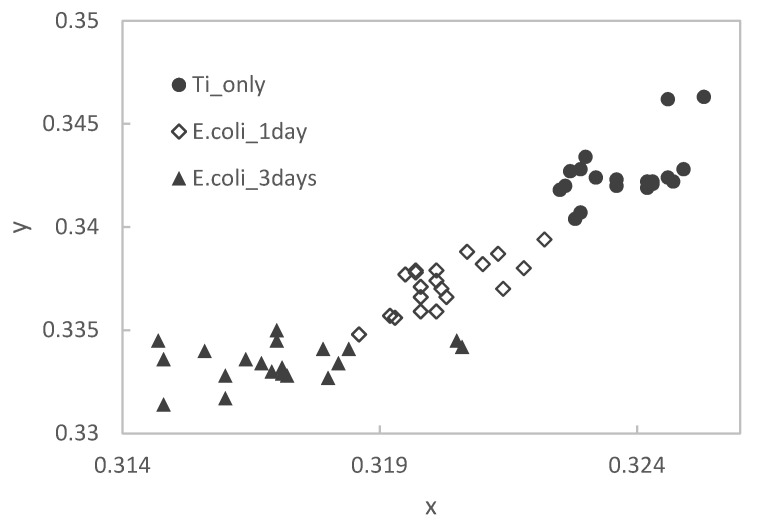
Color changes of stained pure titanium specimens with immersion times.

**Figure 6 materials-15-06727-f006:**
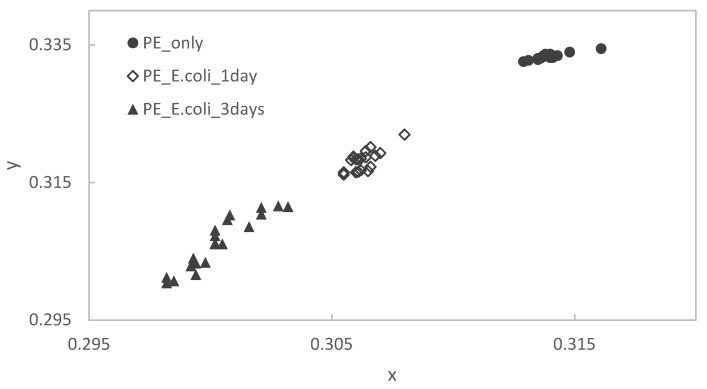
Color changes of stained PE specimens with immersion times.

**Figure 7 materials-15-06727-f007:**
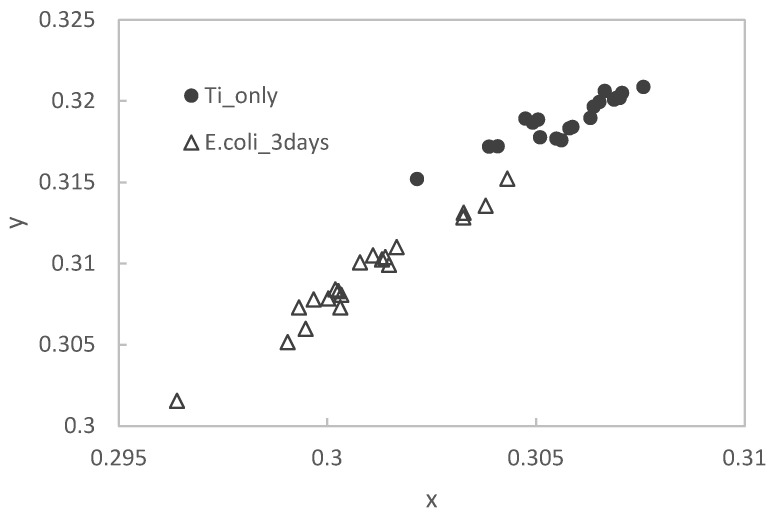
Color changes based on image analyses and calculations.

**Table 1 materials-15-06727-t001:** Average values and their standard deviations for stained titanium specimens.

	Average	Standard Deviation
Ti only (x,y)	(0.3237, 0.3425)	(0.0009, 0.0014)
*E. coli* on Ti for 3 days (x,y)	(0.3171, 0.3335)	(0.0016, 0.0009)

**Table 2 materials-15-06727-t002:** Color difference between titanium and PE specimens.

	Ti	PE
Color Difference ΔC	0.01126	0.03037

**Table 3 materials-15-06727-t003:** Average values and their standard deviations based on image analyses and calculations.

	Average	Standard Deviation
Ti only (x,y)	(0.30670, 0.31886)	(0.00132, 0.00146)
*E. coli* on Ti for 3 days (x,y)	(0.30087, 0.330926)	(0.00184, 0.00317)

## Data Availability

Not applicable.

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
