# Peer review of "Quantitative Analyses of Biofilm by Using Crystal Violet Staining and Optical Reflection"

_materials, 2022, doi:10.3390/ma15196727_

Round 1
Reviewer 1 Report
There are many concerns about this article as presented. The main issues are:
- Lack of clarity regarding the work's objective, methods, and conclusions.
- There is no clear connection between the introduction, the methods, and the conclusions.
- Biofilm's EPS component has been extensively studied, and the consensus is that vary across species. Also, environmental conditions and time play a significant role in EPS composition. Yet, in the point 2 of the conclusions section (Lines 251-252), the authors state that (according to the Raman spectroscopy data) “…the components of BFs formed and their composition ratios are the same regardless of the type of substrate and the type of bacteria…”. As the author's conclusions contradict current knowledge, they must include additional (biochemical) characterization to support their statements that components of BFs are the same between these two bacterial species.
- As presented, these results suggest that Raman spectroscopy cannot distinguish the EPS components and the amounts produced.
Author Response
Dear esteemed Editors and Reviewers:
Thank you so much for your kind reviewing and many useful tips. And please forgive us for the revision delay. The main reason should be attributed mainly to the corresponding author (Hideyuki Kanematsu) and some inevitable reasons. However, we did our best to revise the manuscript, accepting and utilizing your great tips as much as possible. Concretely speaking, our revision actions are described in the attached file.
That’s all for our authors’ action. All of our revision according to the statements above were highlighted in yellow color. The revisions of English expression were also highlighted in blue color.
We did our best to revise the original manuscript. Hopefully, our revision would be sufficiently qualified for the publication of materials’ paper.
Hideyuki Kanematsu on behalf of authors’ team.

Reviewer 2 Report
This work described the analysis of biofilm by CV staining using Raman spectroscopy. The method is helpful from the industrial viewpoint, while I feel it lack of novelty in the research field. I recommend a major revision, and the following issues should be addressed.
- There have been lots of reports on Raman analysis of biofilms. The description on those reports is too rough to understand the background. Ref. "[8-29]" are cited for one sentence. Is it not wield???
- What is the purpose of using Fig. 5? Can the author measure the thickness of the biofilm obtained at different stages?
- The conclusions should be re-considered based on the investigations of the relevant previous reports.
- For Fig.7 and Fig. 8, what the x and y axis? Their meaning should be clearly described in the figures.
- What does "EPS" mean in abstract? Abbreviations should be properly used throughout the manuscript.
- The significance of this work is not clearly presented. I recommend a re-writing of the introduction and conclusion parts.
Author Response

(The authors gave the same response as above.)

Reviewer 3 Report
I appreciate that the authors submitted their research article to MDPI materials. I have read the manuscript precisely and the comments are as follow.
1. In introduction, the authors explained the aim of the study. For comparison between evaluation assays, it would be better to compare with gold standard method. I am not sure that Raman spectroscopy is the best method for evaluating BF. The author should describe the reason that they used Raman spectroscopy.
2. There are several typos. (e.g. - optical refection at keywords)
3. In the graphs, the authors need to add title and unit.
4. In 2-2 Bacteria, the line 70, ‘One of them’ sounds ambiguous. The authors need to name it accurately.
5. In every figure, the authors need to add the caption.
6. In staining data between PE and Ti, the data with PE (Fig 8) looks like distinguishing each group better. Although the authors calculated color difference to compare the substrate, I don’t think that it is a proper method. The authors need to apply another statistical method which can compare their characteristics well.
7. Even though authors insisted that Raman spectroscopy could qualify BF, in Fig 4, the tendency of peaks looks not that different between E.coli and S.epidermidis.
8. In staining experiments, the authors need to show the color images between groups.
9. The paragraphs in discussion need to be located in results.
10. Overall, the aim of this study is not well described.
Author Response

(The authors gave the same response as above.)

Round 2
Reviewer 1 Report
The authors propose a method for quantitative analysis of bacterial biofilms using Crystal violet staining and then analyzing their images.
The applications of the proposed work are unclear, as there are many current methods that provide cost-effective and more precise analytical information on the biofilms traits.
As described, it is unclear what the authors mean by quantitative analysis, as the proposed method does not provide any quantitative information about relevant traits of the biofilm, such as biochemical profile, cell density, or structural organization. Moreover, the work lacks complementary techniques that would support their results.
Author Response
Thank you very much for your precious time and also for your useful advices, tips and suggestions. We decided to accept and reflect them as much as possible. Concrete actions are mentioned below.
You wrote as follows:
------
The authors propose a method for quantitative analysis of bacterial biofilms using Crystal violet staining and then analyzing their images.
The applications of the proposed work are unclear, as there are many current methods that provide cost-effective and more precise analytical information on the biofilms traits.
As described, it is unclear what the authors mean by quantitative analysis, as the proposed method does not provide any quantitative information about relevant traits of the biofilm, such as biochemical profile, cell density, or structural organization. Moreover, the work lacks complementary techniques that would support their results.
-----
We are so sorry that we could not assure you for the core concept of this article and therefore, we might mislead you. However, we added some sentences in the introduction part to improve the contents. And the limitation at this point is also mentioned in the part just before the conclusion part. Concretely speaking, the revised parts are, from line 56 to line 84 in the new version and also from line 236 to 241.
Reviewer 2 Report
The paper can be accepted in current form
Author Response
Dear Dr. Reviewer 2:
Thank you very much for your precious time and useful suggestions. We appreciate. You wrote:
--------
The paper can be accepted in current form.
----
Thank you very much for your positive review and understanding.
Reviewer 3 Report
The comments that I mentioned are well edited.
Author Response
Dear Dr. Reviewer 3:
thank you very much for your precious time and useful suggestions.
You wrote this time:
------
The comments that I mentioned are well edited.
-----
Thank you so much for your positive understanding and evaluation.